# Observer-based prescribed-time lag bipartite consensus of nonlinear multi-agent systems under event-triggered mechanism

Jialong Tian[1], Tao Li [1]*, Xiaowen Zhao[2], Yuanmei Wang[3], Haiyang Hua[1], Le Yan[1]

**1** School of Electrical Engineering and Automation, Hubei Normal University, Huangshi, China, **2** School of Mathematics, Hefei University of Technology, Hefei, China, **3** School of Electronic Information and Electrical Engineering, Yangtze University, Jingzhou, China

* taohust2008@163.com

## Abstract

Considering the complexity of lag time and the convergence time, as well as the influence of the observer and the event-triggered mechanism, the prescribed-time lag bipartite consensus (PTLBC) control problem for nonlinear multi-agent systems (MASs) is researched in the paper. First of all, it designs the prescribed-time dynamic observer (PTDO) for followers to get the states of the leader within an arbitrarily prescribed time. Furthermore, to significantly decrease the communication consumption, the innovative event-triggered mechanism is studied for followers. To realize the lag bipartite consensus (LBC) of nonlinear MASs within an arbitrarily prescribed time, a PTLBC control scheme is presented via the aforementioned PTDO and event-triggered mechanism. With Lyapunov stability analysis method, sufficient conditions are obtained and detailed stability analysis is studied, which indicates that the nonlinear MASs can realize PTLBC. Furthermore, the analysis proves that the proposed event-triggered mechanism excludes Zeno behaviour. The theoretical analysis is validated by means of a simulation example.

## 1. Introduction

Recently, the cooperative control of MASs has been extensively studied across numerous scientific fields, including urban transportation scheduling, distributed power generation, and unmanned aerial vehicles formation control, etc.[1–8]. As a primary problem in the cooperative control of MASs, consensus control has achieved many significant results [9–15].

Lag consensus, as an unusual case of consensus, has recently attracted considerable attention for its utilisation in preventing congestion [16–19]. Assuming to be a lag time, lag consensus denotes that the followers' states at time are same as the leader's states at time. The asymptotic stability of lag consensus has been studied for MASs [16–19]. However, the asymptotic lag consensus cannot satisfy

**Data availability statement:** All relevant data are within the manuscript and its Supporting Information files.

**Funding:** This work is supported by the National Natural Science Foundation of China under Grants 62473135 and 62173121, in part by the Natural Science Foundation of Hubei Province of China under Grant 2024AFB812.

**Competing interests:** The authors have declared that no competing interests exist.

the requirements of practical engineering applications. Finite-time lag consensus (FNTLC) control has recently gained attention for enhancing the convergence speed of the MASs [20–22]. However, FNTLC control is subject to initial states of MASs. For resolving the problem, fixed-time lag consensus (FXTLC) control has been presented [23–26]. But FXTLC control depends upon the control protocol parameters. To address the issue, prescribed-time lag consensus (PTLC) control has been proposed. The prescribed-time lag consensus control protocol makes the arbitrary assignment of convergence time possible, which is independent of both the controller parameters and the initial states of the agents. Obviously, PTLC has important academic value and practical significance.

The problem of prescribed-time partial component lag consensus of MASs subject to both disturbances and switching topologies is investigated [27]. A PTLC control scheme with adaptive gains for lag consensus is introduced [28]. It should be noted that the above works, including [27,28], all assume that the communications among agents are continuous. However, continuous communication results in excessive resource consumption and may compromise the system's stability due to the limited computational capabilities and storage capacity in applications. Obviously, this research of discontinuous communication control protocols is important.

To reduce resource consumption, a control scheme for successive lag consensus of nonlinear MASs is presented based on event-triggered mechanism with both constant and time-varying lag consensus scenarios [29,30]. By a novel static event-triggered mechanism, reference [31] designs a control protocol to research the lag consensus of MASs. Reference [1] develops a dynamic memory event-triggered mechanism to research the lag consensus control of multi-unmanned aerial vehicle systems under hybrid attacks. The aforementioned studies have effectively addressed the resource constraint of agents in practical scenarios using event-triggered mechanisms, but they rarely consider the convergence time at the same time. It is more challenging to create a control scheme for reducing resource consumption while setting the prescribed convergence time at will for the lag consensus control problem.

Notably, the aforementioned works about lag consensus have primarily considered cooperative relationships while neglecting competitive relationships within agents. However, competitive relationship is widely present in natural systems, social and engineering systems. In MASs, cooperation and competition often coexist. Motivated by competitive relationships and lag consensus, the problem of lag bipartite consensus is presented [32–35]. Reference [34] investigates the LBC control problem of linear MASs under saturating input, but utilizes continuous communication control protocol. Using the event-triggered mechanism, reference [35] discusses the LBC control of nonlinear MASs with external disturbances, but just considers the asymptotic convergence.

Significantly, the aforementioned studies on LBC assume that the leader's states are available. Obviously, it is unrealistic in numerous practical applications. So, to devise an observer is important for estimating the states of leader. In light of the discussions above, the paper comprehensively studies the PTLBC of nonlinear MASs via the PTDO and event-triggered mechanism. The main contributions of this paper are summarized as follows:

1) Different from the system investigated in [36], this paper studies a second-order integrator system with nonlinear components, which extends the research scope of relevant works.

2) Different from the method adopted in [28], this paper proposes a novel event-triggered mechanism by combining the prescribed-time time-varying function with the measurement error.

3) Considering that follower agents cannot directly access the state of the leader agent, a prescribed-time dynamic observer is designed to estimate the leader's state information.

4) To address the problem of communication congestion, a prescribed-time lag-bipartite consensus control protocol based on the event-triggered mechanism is designed by integrating prescribed-time control with the event-triggered mechanism.

This paper is organized into the following chapters: The problem statement and preliminaries are shown in Section 2. Section 3 proposes a novel PTDO, studies an innovative event-triggered mechanism, develops a PTLBC control protocol to realize LBC within the arbitrarily prescribed time, and discusses the Zeno behaviour in detail. The theoretical results are validated through a numerical example in Section 4. Section 5 gives the conclusions.

## 2. Preliminaries and problem statement

### 2.1. Graph theory

Let $\mathbb{R}^q$ be the set of the $q \times 1$ real vectors, $I_q$ be the $q$ order identity matrix, and $\mathbb{R}^{q \times q}$ be the set of $q \times q$ real matrices. $sign(\cdot)$ is the sign function. $\|\cdot\|$ is the Euclidean norm. The binary operator $\otimes$ represents the Kronecker product. $\lambda_{\min}(H)$ and $\lambda_{\max}(H)$ denote the minimum and maximum eigenvalues of the symmetric matrix $H$, separately.

The followers' connected topology of the nonlinear MASs is typically represented with a directed signed graph $\mathcal{G} = \{\mathcal{V}, \mathcal{E}, \mathcal{A}\}$. Let $\mathcal{V} = \{v_1, ..., v_n\}$ be the set of followers, $\mathcal{E} \subseteq \mathcal{V} \times \mathcal{V}$ be the set of edges, and $\mathcal{A} = [a_{ij}] \in \mathbb{R}^{n \times n}$ be the adjacency matrix. If $\mathcal{E}_{ij} \in \mathcal{E}$, then $a_{ij} \neq 0$, otherwise, $a_{ij} = 0$. $a_{ij} < 0$ indicates the competitive relationship between the $i$-th follower and $j$-th follower, while $a_{ij} > 0$ represents the cooperative relationship between the $i$-th follower and $j$-th follower. $L = [l_{ij}] \in \mathbb{R}^{n \times n}$ is the Laplacian matrix of $\mathcal{G}$, satisfying $l_{ij} = -a_{ij}$ for $i = j$, and $l_{ij} = \sum_{j=1}^{N} |a_{ij}|$ for $i \neq j$.

Furthermore, the interaction directed signed graph $\tilde{\mathcal{G}} = \{\tilde{\mathcal{V}}, \tilde{\mathcal{E}}, \tilde{\mathcal{A}}\}$ describes the connected topology of the whole system. Let $\mathcal{V}_0 = \{v_0\}$ be the leader's set. Define $\tilde{\mathcal{V}} = \mathcal{V} \cup \mathcal{V}_0$. The set $\tilde{\mathcal{V}}$ is partitioned into a pair of disjoint subsets $\tilde{\mathcal{V}}_1$ and $\tilde{\mathcal{V}}_2$. The matrix $B = diag(b_1, b_2, ..., b_n)$ describes the relationship between the followers and the leader. When there is a connection from the leader to $i$-th follower, $b_i = 1$; otherwise, $b_i = 0$.

### 2.2. Some assumptions and lemmas

The section lists the useful assumptions and lemmas.

**Assumption 1 ([26]).** Let the graph $\mathcal{G}$ be structurally balanced. There is a spanning tree rooted at the leader in the graph $\tilde{\mathcal{G}}$.

**Assumption 2 ([37,38]).** There always exist positive real numbers $c_x$ and $c_v$ satisfying the below equation, where vectors $n, m, \overline{n}, \overline{m} \in \mathbb{R}^q$.

$$\|f(n, m, t) - f(\overline{n}, \overline{m}, t)\| = c_x\|n - \overline{n}\| + c_v\|m - \overline{m}\|.$$

**Lemma 1 ([14]).** If there is a matrix $\theta = diag\{\sigma_1, \sigma_2, ..., \sigma_n\}$, where $\sigma_i \in \{1, -1\}$, and all elements of $\tilde{\mathcal{A}} = \theta^T \mathcal{A}\theta$ are greater than or equal to zero, where $\theta^T = \theta^{-1} = \theta$, then the structural balance of the directed signed graph $\mathcal{G}$ holds.

                                                          

**Lemma 2 ([19]).** The below inequality is true

$$\|y\|\|z\| \leq l\|y\|^2 + \frac{1}{4l}\|z\|^2,$$

where $y$, $z$ are any given vectors with proper dimensions and $l$ is a positive constant.

**Lemma 3 ([26]).** If there is a positive define matrix $\Xi = diag\,(\xi_i) = diag\,(x_i/y_i)$, then $Q = +H^T\Xi$ is positive define, where the vectors $y_i = [y_1, y_2,\ldots,y_n]^T = H^{-T}1_n, x_i = [x_1, x_2, \ldots, x_n]^T = H^{-1}1_n$ and $H = L + B$.

**Lemma 4 ([26]).** Analyze the following systems

$$\dot{z}\,(t) = f\,(t, z\,(t))\,, \tag{1}$$

where $z\,(t) \in \mathbb{R}^q$ is the state, $f(\cdot, \cdot)$ is the time-bounded vector and $f\,(0)$ indicates the system initial state. The time-varying function below is first introduced [36].

$$\psi\,(t) = \left(\frac{T}{T+t_0-t}\right)^\rho,\quad t_0 \leq t < t_0 + T, \tag{2}$$

and $\psi\,(t) = 1$, $t \geq t_0 + T$, in which $\rho > 1$, $T > 0$ is a prescribed time, and $t_0 > 0$ is the initial time. $\Psi\,(t)$ is defined as

$$\Psi\,(t) = \begin{cases} \frac{\dot{\psi}\,(t)}{\psi\,(t)}, & t_0 \leq t < t_0 + T, \\ \frac{h}{T}, & t \geq t_0 + T. \end{cases} \tag{3}$$

For system, if there is a Lyapunov function $V\,(t)$ which satisfies

$$\dot{V}\,(t) + cV\,(t) + k\Psi\,(t)\,V\,(t) \leq 0,\quad t_0 \leq t < t_0 + T\,, \tag{4}$$

where $\Psi\,(t)$ is defined in, $c \geq 0$ and $k > 0$, then the system is prescribed-time stable with the prescribed time $t_0 + T$. It can get the solution below.

$$\begin{cases} V\,(t) \leq \psi(t)^{-k}e^{-c(t-t_0)}\,V\,(t_0)\,, & t_0 \leq t < t_0 + T, \\ V\,(t) = 0, & t \geq t_0 + T. \end{cases} \tag{5}$$

**Lemma 5 ([38]).** For system, if there exist a continuously differentiable and positive-definite function $V\,(t)$, two positive constants $n_1 > 0$, $n_2 > 0$, and a scalar $n_3$ satisfying $(1/h) - n_1 < n_3 < (1/h)$, such that

$$\begin{cases} \dot{V}\,(t) \leq -n_1\Psi\,(t)\,V\,(t) + n_2\psi^{-n_3}\,(t)\,, & t_0 \leq t < t_0 + T, \\ \dot{V}\,(t) \leq -n_1\Psi\,(t)\,V\,(t)\,, & t \geq t_0 + T, \end{cases} \tag{6}$$

then the system can realize the prescribed-time stability. It gets the solution as

$$\begin{cases} V\,(t) \leq \left[e^{-n_1(t-t_0)}\,V\,(t_0) + \frac{n_2 T}{\rho\,(n_1+n_3)-1}\right]\psi^{n_3-\frac{1}{\rho}}\,(t)\,, & t_0 \leq t < t_0 + T, \\ V\,(t) = 0, & t \geq t_0 + T. \end{cases} \tag{7}$$

## 2.3. Problem statement

Analyze the nonlinear MASs system of $n$ followers and one leader. The leader's dynamics is modelled below.

$$\begin{cases} \dot{x}_0\left(t\right) = v_0\left(t\right), \\ \dot{v}_0\left(t\right) = u_0\left(t\right) + g\left(t, x_0\left(t\right), v_0\left(t\right)\right), \end{cases} \tag{8}$$

where $x_0\left(t\right) \in \mathbb{R}^r$ represents the position, $v_0\left(t\right) \in \mathbb{R}^r$ represents the velocity, $g\left(t, x_0\left(t\right), v_0\left(t\right)\right) \in \mathbb{R}^r$ represents nonlinear functions, and $u_0\left(t\right) \in \mathbb{R}^r$ represents the acceleration or the control input.

The $i$-th follower's nonlinear dynamics is modelled below.

$$\begin{cases} \dot{x}_i\left(t\right) = v_i\left(t\right), \\ \dot{v}_i\left(t\right) = u_i\left(t\right) + g\left(t, x_i\left(t\right), v_i\left(t\right)\right), \end{cases} \tag{9}$$

where $x_i\left(t\right) \in \mathbb{R}^r$ represents the position, $v_i\left(t\right) \in \mathbb{R}^r$ represents the velocity, $g\left(t, x_i\left(t\right), v_i\left(t\right)\right) \in \mathbb{R}^r$ indicates nonlinear functions satisfying Lipschitz condition, and $u_i\left(t\right) \in \mathbb{R}^r$ indicates the control input.

**Definition 1.** For the lag time $\tau$ and the arbitrarily prescribed time $T > 0$, if the solution of and satisfies

$$\begin{aligned} \lim_{t \to t_0 + T} \|x_i\left(t\right) - \sigma_i x_0(t - \tau)\| = 0, \\ \lim_{t \to t_0 + T} \|v_i\left(t\right) - \sigma_i v_0(t - \tau)\| = 0, \end{aligned} \tag{10}$$

and

$$\begin{aligned} \|x_i\left(t\right) - \sigma_i x_0(t - \tau)\| = 0, \\ \|v_i\left(t\right) - \sigma_i v_0(t - \tau)\| = 0, \qquad t \geq t_0 + T, \end{aligned} \tag{11}$$

then the PTLBC is said to be achieved for the nonlinear MASs and, where $\sigma_i$ is defined in Lemma 1.

## 3. Main results

### 3.1. The design of PTDO

Suppose Assumption 1 holds. For the followers, the PTDO below is designed to estimate the states information of the leader.

$$\begin{cases} \dot{\vartheta}_{vi}\left(t\right) = u_0 - \beta\left(c + k\Psi_1\left(t\right)\right)\xi_{vi}\left(t\right), \\ \dot{\vartheta}_{xi}\left(t\right) = \vartheta_{vi}\left(t\right) - \beta\left(c + k\Psi_2\left(t\right)\right)\xi_{xi}\left(t\right), \end{cases} \tag{12}$$

where

$$\begin{cases} \xi_{xi}\left(t\right) = \sum_{j=1}^{n} |a_{ij}|\left(\vartheta_{xi}\left(t\right) - sign\left(a_{ij}\right)\vartheta_{xj}\left(t\right)\right) + b_i\left(\vartheta_{xi}\left(t\right) - \sigma_i x_0\left(t\right)\right), \\ \xi_{vi}\left(t\right) = \sum_{j=1}^{n} |a_{ij}|\left(\vartheta_{vi}\left(t\right) - sign\left(a_{ij}\right)\vartheta_{vj}\left(t\right)\right) + b_i\left(\vartheta_{vi}\left(t\right) - \sigma_i v_0\left(t\right)\right). \end{cases}$$

Here, the estimate of the leader state $x_0$ and $v_0$ are denoted as $\vartheta_{xi}$ and $\vartheta_{vi}$, respectively. $\beta$ is the observer gains, $\Psi_\iota(t)$ is as follows

$$\Psi_\iota(t) = \begin{cases} \frac{\dot{\psi}_\iota(t)}{\psi_\iota(t)}, & t_0 \le t < t_0 + \iota T, \\ \frac{h}{\iota T}, & t \ge t_0 + \iota T, \end{cases}$$

(13)

where $\psi_\iota(t) = \left(\frac{\iota T}{\iota T + t_0 - t}\right)^\rho$, $t \ge t_0 + \iota T$ and $\iota$ ($\forall \iota = 1, 2, 3$). Set three times $T_1$, $T_2$, and $T_3$, where $T_1 = t_0 + T$ and $T_2 = t_0 + 2T$ represent the prescribed convergence time of the position states and velocity states for the state observer, respectively; $T_3 = t_0 + 3T$ denotes the prescribed convergence time of the closed-loop nonlinear system under the proposed controller.

**Theorem 1.** For the PTDO, estimation errors are categorised into velocity errors $\overline{\vartheta}_{vi}(t) = \vartheta_{vi}(t) - \sigma_i v_0(t)$ and position errors $\overline{\vartheta}_{xi}(t) = \vartheta_{xi}(t) - \sigma_i x_0(t)$. If $\beta > \lambda_{\max}(\Xi)/\lambda_{\min}(Q)$, $c \ge 0$, $k > 0$ and $h > 1$, then $\overline{\vartheta}_{xi}$ and $\overline{\vartheta}_{vi}$ will converge to zero within the prescribed time $T_1 = t_0 + T$ and $T_2 = t_0 + 2T$, respectively.

**Proof.** The convergence analysis of prescribed time dynamic observer will be implemented by the following two steps.

(1) For convenience, let $\vartheta_{xi}(t)$ be $\vartheta_{xi}$, and $\vartheta_{vi}(t)$ be $\vartheta_{vi}$. Considering the observer error $\overline{\vartheta}_{vi}$, by differentiating the first equation in, we obtain $\dot{\overline{\vartheta}}_v = -\beta(k + c\Psi_1(t))(H \otimes I_r)\overline{\vartheta}_v$. Selecting the following Lyapunov function

$$V_1 = \frac{1}{2}\overline{\vartheta}_v^T(\Xi \otimes I_r)\overline{\vartheta}_v,$$

(14)

the derivative of $V_1$ is as follows

$$\begin{aligned} \dot{V}_1 &= -\beta(k + c\Psi_1(t))\overline{\vartheta}_v^T(\Xi H \otimes I_r)\overline{\vartheta} \\ &= -\frac{1}{2}\beta(k + c\Psi_1(t))\overline{\vartheta}_v^T(Q \otimes I_r)\overline{\vartheta}_v \\ &\le \beta(k + c\Psi_1(t))\lambda_{\min}(Q)\overline{\vartheta}_v^T\overline{\vartheta}_v. \end{aligned}$$

(15)

From (14), it can be obtained that $-1/2\lambda_{\max}(\Xi)\overline{\vartheta}_v^T\overline{\vartheta}_v \le -V_1 \le -1/2\lambda_{\min}(\Xi)\overline{\vartheta}_v^T\overline{\vartheta}_v$. It concludes that $-\overline{\vartheta}_v^T\overline{\vartheta}_v \le -2/\lambda_{\max}(\Xi)V_1$. Then, substituting these inequalities into, it obtains

$$\begin{aligned} \dot{V}_1 &\le -\beta(k + c\Psi_1(t))\frac{\lambda_{\min}(Q)}{\lambda_{\max}(\Xi)}V_1 \\ &\le -(k + c\Psi_1(t))V_1, \end{aligned}$$

(16)

where $-\beta\lambda_{\min}(Q)/\lambda_{\max}(\Xi) \le -1$, i.e., $\beta \ge \lambda_{\max}(\Xi)/\lambda_{\min}(Q)$. Especially, the following solution can be obtained

$$\begin{cases} V_1 \le \psi(t)^{-k}e^{-c(t-t_0)}V(t_0), & t \in [t_0, t_0 + T_1), \\ V_1 \equiv 0, & t \in [t_0 + T_1, \infty). \end{cases}$$

In other words, within the arbitrarily prescribed time $T_1$, $\overline{\vartheta}_{vi}$ converges to zero according to Lemma 4.

(2) Similar to the derivation processes of Equations (14) and (15) and, it obtains that within the arbitrarily prescribed time $T_2$, $\overline{\vartheta}_{xi}$ will converge to zero. That is to say, the PTDO (12) is effective.

## 3.2. Event-triggered mechanism

The innovative event-triggered mechanism is devised in the section. For simplicity, let $x_0(t-\tau)$ be $x_0^\tau(t)$, $v_0(t-\tau)$ be $v_0^\tau(t)$, $\vartheta_{xi}(t-\tau)$ be $\vartheta_{xi}^\tau(t)$, and $\vartheta_{vi}(t-\tau)$ be $\vartheta_{vi}^\tau(t)$. Define the lag bipartite consensus error as $\bar{x}_i(t) = x_i(t) - \sigma_i x_0^\tau(t)$ and $\bar{v}_i(t) = v_i(t) - \sigma_i v_0^\tau(t)$, respectively. Let two auxiliary states be

$$\widehat{x}_i(t) = \tilde{\Psi}_3(t)\,\bar{x}_i(t),$$
$$\widehat{v}_i(t) = \bar{v}_i(t). \tag{17}$$

Then, let $\eta_i(t) = k_1\widehat{x}_i(t) + k_2\widehat{v}_i(t)$, where $k_1 > 0$, $k_2 > 0$. Let $\tilde{\Psi}_3(t) = \dot{\Psi}_3(t)/\Psi_3(t)$ for $t_0 \le t < T_3$, and $\tilde{\Psi}_3(t) = 0$ for $t \ge T_3$. Define the state measured error $e_i(t) = \eta_i(t_k^i) - \eta_i(t)$. Denote $e(t) = [e_1(t), e_2(t), ..., e_n(t)]^T$. Choose the event-triggered function as follows

$$h_i(t) = \|e_i(t)\| - \alpha\|\eta_i(t)\| - \gamma_i\psi^{-2b}(t), \tag{18}$$

where $b > 0$, $\alpha > 0$, $\gamma_i > 0$ are positive constants, and $\psi(t)$ defined in (2). If $h_i(t) \ge 0$, the position $x_i(t_k^i)$ and velocity $v_i(t_k^i)$ are update to $x_i(t_{k+1}^i)$ and $v_i(t_{k+1}^i)$, and transmitted to the MASs, where

$$t_{k+1}^i = \inf\{t|t > t_k^i : h_i(t) \ge 0\}, \qquad k \in \mathbb{N}, \quad i = 1, 2, ..., N.$$

## 3.3. PTLBC control protocol

Via the above event-triggered mechanism and PTDO, the PTLBC control protocol is designed. For $t_0 \le t < T_2$, we design the $i$-th follower's PTLBC control protocol as follows

$$
u_i(t) = -\Psi_3(t)\left[k_1\Psi_3(t)\left(\sum_{j=1}^{N}|a_{ij}|\left(x_i(t_k^i) - \text{sgn}(a_{ij})x_j(t_k^j)\right) + b_i\left(x_i(t_k^i) - \vartheta_{xi}^\tau(t_k^i)\right)\right)\right.
$$
$$
\left. + k_2\left(\sum_{j=1}^{N}|a_{ij}|\left(v_i(t_k^i) - \text{sgn}(a_{ij})v_j(t_k^j)\right) + b_i\left(v_i(t_k^i) - \vartheta_{vi}^\tau(t_k^i)\right)\right)\right], \qquad t \in [t_k^i, t_{k+1}^i) \tag{19}
$$

where $k_1, k_2 > 0$ and $\Psi_3(t)$ corresponds to the case when $\iota = 3$ in (3). Besides, $t_k^i$ represents the triggering time of the $k$-th event for the follower $i$, where $t_0^i = 0$. For $t \in [t_k^i, t_{k+1}^i)$, $x_i(t_k^i)$ and $v_i(t_k^i)$ record the transmission states of follower $i$ at $t_k^i$. When $t \ge T_2$, the $\vartheta_{xi}^\tau(t_k^i)$ and $\vartheta_{vi}^\tau(t_k^i)$ transition to $x_0^\tau(t_k^i)$ and $v_0^\tau(t_k^i)$ in the control protocol (19). Then, it obtains

$$
u_i(t) = -\Psi_3(t)\left[k_1\Psi_3(t)\left(\sum_{j=1}^{N}|a_{ij}|\left(x_i(t_k^i) - \text{sgn}(a_{ij})x_j(t_k^j)\right) + b_i\left(x_i(t_k^i) - \sigma_i x_0^\tau(t_k^i)\right)\right)\right.
$$
$$
\left. k_2\left(\sum_{j=1}^{N}|a_{ij}|\left(v_i(t_k^i) - \text{sgn}(a_{ij})v_j(t_k^j)\right) + b_i\left(v_i(t_k^i) - \sigma_i v_0^\tau(t_k^i)\right)\right)\right], \qquad t \in [t_k^i, t_{k+1}^i) \tag{20}
$$

Taking the derivative of $\widehat{x}_i(t)$, one gets

$$\dot{\widehat{x}}_i(t) = \tilde{\Psi}_3(t)\,\bar{x}_i(t) + \Psi_3(t)\,\widehat{v}_i(t), \tag{21}$$

Consequently, we obtain the feedback control system of (8) and (9) together with (19) and (21) as

$$
\begin{cases}
\dot{\widehat{x}}_i(t) = \tilde{\Psi}_3(t)\,\widehat{x}_i(t) + \Psi_3(t)\,\widehat{v}_i(t), \\
\dot{\widehat{v}}_i(t) = -k_1\Psi_3(t)\displaystyle\sum_{j=1}^{N} l_{ij}\widehat{x}_i(t_k^i) - k_2\Psi_3(t)\displaystyle\sum_{j=1}^{N} l_{ij}\widehat{v}_i(t_k^i) + g\left(t, x_i(t), v_i(t)\right) - g\left(t, x_0^\tau(t), v_0^\tau(t)\right).
\end{cases}
$$
(22)

The system (22) has the concise form below,

$$
\begin{cases}
\dot{\widehat{x}}(t) = \tilde{\Psi}_3(t)\,\widehat{x}(t) + \Psi_3(t)\,\widehat{v}(t), \\
\dot{\widehat{v}}(t) = -\Psi_3(t)\left(H \otimes I_m\right)\left(k_1\widehat{x}(t_k^i) + k_2\widehat{v}(t_k^i)\right) + \mathcal{F}(t).
\end{cases}
$$
(23)

where $\mathcal{F}(t) = G(t, x, v) - 1_n \otimes g(t, x_0^\tau, v_0^\tau)$, $G(t, x, v) = \left[g^T(t, x_1, v_1), g^T(t, x_2, v_2), \ldots, g^T(t, x_N, v_N)\right]^T$,
$\widehat{x} = \left[\widehat{x}_1^T, \widehat{x}_2^T, \ldots, \widehat{x}_N^T\right]^T$, $\widehat{v} = \left[\widehat{v}_1^T, \widehat{v}_2^T, \ldots, \widehat{v}_N^T\right]^T$, $x = \left[x_1^T, x_2^T, \ldots, x_N^T\right]^T$, and $v = \left[v_1^T, v_2^T, \ldots, v_N^T\right]^T$.

Substituting state measurement error $e(t)$ into (23), it has

$$
\begin{cases}
\dot{\widehat{x}}(t) = \tilde{\Psi}_3(t)\,\widehat{x}(t) + \Psi_3(t)\,\widehat{v}(t), \\
\dot{\widehat{v}}(t) = -\Psi_3(t)\left(H \otimes I_m\right)\left(\eta(t) + e(t)\right) + \mathcal{F}(t).
\end{cases}
$$
(24)

**Theorem 2.** Assume that Assumptions 1 and 2 hold. Consider the MASs (8) and (9) with the PTLBC control protocol (19). If the following conditions can be satisfied,

$$
\begin{aligned}
&k_1 \upsilon_2 - k_2^2 < 0, \qquad \Delta_1 + \Delta_3 < 0, \qquad \Delta_2 + \Delta_4 < 0, \\
&\Delta_1 = \left(\frac{k_1}{h} - \frac{k_1^2}{2k_2}\right)\frac{\lambda_{max}(Q)}{\lambda_{min}(Q)} + \frac{k_1}{2k_2 h}\upsilon_1, \\
&\Delta_2 = \frac{k_1}{k_2}\upsilon_1\left(1 + \frac{1}{\rho}\right) - \frac{k_2}{2}, \\
&\Delta_3 = \upsilon_2\left(\frac{k_1 Tc_x}{k_2 h} + \frac{Tlc_x}{h} + \frac{k_1 lc_v}{k_2}\right) + \frac{lk_1}{2k_2} + \frac{(k_1 + k_2)\lambda_{max}(Q)}{8k_2 l\lambda_{min}(Q)}\alpha^2 k_1^2, \\
&\Delta_4 = \upsilon_2\left(c_v + \frac{Tc_x}{h4l} + \frac{k_1 c_v}{k_2 4l}\right) + \frac{1}{4l} + \frac{(k_1 + k_2)\lambda_{max}(Q)}{8k_2 l\lambda_{min}(Q)}\alpha^2 k_2^2.
\end{aligned}
$$
(25)

where $\upsilon_1 = \frac{\lambda_{max}(\Xi)}{\lambda_{max}(Q)}$, $\upsilon_2 = \frac{\lambda_{max}(\Xi)}{\lambda_{min}(Q)}$, and definition $\Delta_5 = \lambda_{max}(Q)\gamma_i^2$. Then the lag bipartite consensus errors $\bar{x}_i(t)$ and $\bar{v}_i(t)$ converge to 0 within arbitrarily prescribed time $t_0 + 3T$, i.e., using the proposed control protocol (19), the PTLBC of the MASs in (8) and (9) can realized.

**Proof.** Let $\delta = \left[\tilde{x}^T, \tilde{v}^T\right]^T$. It can choose the Lyapunov function below,

$$
V_2 = \frac{1}{2}\delta^T\left(\Omega \otimes I_r\right)\delta,
$$
(26)

where

$$
\Omega = \begin{bmatrix} k_1 Q & \frac{k_1}{k_2}\Xi \\ \frac{k_1}{k_2}\Xi & \Xi \end{bmatrix}.
$$

Here, $Q$ and $\Xi$ are explained in Lemma 3. It can get that $\Xi$ is positive define based on Lemma 2. Using the first inequality of (25), it gets $V_2 \geq 0$. Then

$$\dot{V}_2 = \frac{k_1}{h}\Psi_3\left(t\right)\tilde{x}^T\left(Q\otimes I_r\right)\tilde{x} - \frac{k_1^2}{k_2}\Psi_3\left(t\right)\tilde{x}^T\left(H\Xi\otimes I_r\right)\tilde{x} + \frac{k_1}{k_2}\Psi_3\left(t\right)\tilde{v}^T\left(\Xi\otimes I_r\right)\tilde{v} - k_2\Psi_3\left(t\right)\tilde{v}^T\left(H\Xi\otimes I_r\right)\tilde{v}$$
$$+\tilde{x}^T\left(k_1\Psi_3\left(t\right)\left(Q\otimes I_r\right) + \frac{k_1}{k_2\rho}\Psi_3\left(t\right)\left(\Xi\otimes I_r\right)\right)\tilde{v} - k_1\Psi_3\left(t\right)\tilde{x}^T\left(Q\otimes I_r\right)\tilde{v} - \left(\frac{k_1}{k_2}\tilde{x}^T + \tilde{v}^T\right)\Psi_3\left(t\right)\left(H\Xi\otimes I_r\right)e\left(t\right)$$
$$+\left(\frac{k_1}{k_2}\tilde{x}^T + \tilde{v}^T\right)\left(\Xi\otimes I_r\right)\mathcal{F}\left(t\right).$$

(27)

Based on Assumption 2, Lemma 3 and $\frac{1}{\Psi_3\left(t\right)} \le \frac{3T}{\rho}$, one gets

$$\left(\frac{k_1}{k_2}\tilde{x}^T + \tilde{v}^T\right)\left(\Xi\otimes I_r\right)\mathcal{F}\left(t\right) = \sum_{i=1}^N \xi_i\left(\frac{k_1}{k_2}\tilde{x}_i^T + \tilde{v}_i^T\right)\left[g\left(t, x_i, v_i\right) - g\left(t, x_0^\tau, v_0^\tau\right)\right]$$
$$\le \sum_{i=1}^N \xi_i\left[\frac{k_1 c_x}{k_2\Psi\left(t\right)}\|\tilde{x}_i\|^2 + c_v\|\tilde{v}_i\|^2 + \left(\frac{c_x}{\Psi\left(t\right)} + \frac{k_1 c_v}{k_2}\right)\|\tilde{x}_i\|\|\tilde{v}_i\|\right]$$
$$\le \left(\frac{k_1 Tc_x}{k_2\rho} + \frac{Tlc_x}{\rho} + \frac{k_1 lc_v}{k_2}\right)\tilde{x}^T\left(\Xi\otimes I_r\right)\tilde{x} + \left(c_v + \frac{Tc_x}{\rho 4l} + \frac{k_1 c_v}{k_2 4l}\right)\tilde{v}^T\left(\Xi\otimes I_r\right)\tilde{v}$$
$$\le \upsilon_2\left(\frac{k_1 Tc_x}{k_2\rho} + \frac{Tlc_x}{\rho} + \frac{k_1 lc_v}{k_2}\right)\tilde{x}^T\left(Q\otimes I_r\right)\tilde{x} + \upsilon_2\left(c_v + \frac{Tc_x}{\rho 4l} + \frac{k_1 c_v}{k_2 4l}\right)\tilde{v}^T\left(Q\otimes I_r\right)\tilde{v}.$$

(28)

From the trigger condition (18), one gets

$$-\left(\frac{k_1}{k_2}\tilde{x}^T + \tilde{v}^T\right)\Psi_3\left(t\right)\left(H\Xi\otimes I_r\right)e\left(t\right) = -\frac{k_1}{2k_2}\Psi_3\left(t\right)\tilde{x}^T\left(Q\otimes I_r\right)e\left(t\right) - \frac{1}{2}\Psi_3\left(t\right)\tilde{v}^T\left(Q\otimes I_r\right)e\left(t\right)$$
$$\le \frac{k_1}{2k_2}\Psi_3\left(t\right)\left(l\tilde{x}^T\left(Q\otimes I_r\right)\tilde{x} + \frac{1}{4l}e^T\left(t\right)\left(Q\otimes I_r\right)e\left(t\right)\right)$$
$$+\frac{1}{2}\Psi_3\left(t\right)\left(l\tilde{v}^T\left(Q\otimes I_r\right)\tilde{v} + \frac{1}{4l}e^T\left(t\right)\left(Q\otimes I_r\right)e\left(t\right)\right)$$
$$\le \frac{lk_1}{2k_2}\Psi_3\left(t\right)\tilde{x}^T\left(Q\otimes I_r\right)\tilde{x} + \frac{l}{2}\Psi_3\left(t\right)\tilde{v}^T\left(Q\otimes I_r\right)\tilde{v}$$
$$+\frac{k_1+k_2}{8k_2 l}\Psi_3\left(t\right)e^T\left(t\right)\left(Q\otimes I_r\right)e\left(t\right),$$

(29)

where

$$e^T\left(t\right)\left(Q\otimes I_r\right)e\left(t\right) \le \lambda_{\max}\left(Q\right)e^T\left(t\right)e\left(t\right) \le \lambda_{\max}\left(Q\right)\left(\alpha^2\|\eta_i\left(t\right)\|^2 + \gamma_i^2\psi^{-4b}\left(t\right)\right)$$
$$\le \lambda_{\max}\left(Q\right)\alpha^2\|k_1\tilde{x}\left(t\right)\|^2 + \lambda_{\max}\left(Q\right)\alpha^2\|k_2\tilde{v}\left(t\right)\|^2 + \lambda_{\max}\left(Q\right)\gamma_i^2\psi^{-4b}\left(t\right)$$
$$\le \frac{\lambda_{\max}\left(Q\right)}{\lambda_{\min}\left(Q\right)}\alpha^2 k_1^2\tilde{x}^T\left(Q\otimes I_r\right)\tilde{x} + \frac{\lambda_{\max}\left(Q\right)}{\lambda_{\min}\left(Q\right)}\alpha^2 k_2^2\tilde{v}^T\left(Q\otimes I_r\right)\tilde{v}$$
$$+\lambda_{\max}\left(Q\right)\gamma_i^2\psi^{-4b}\left(t\right).$$

(30)

Applying Lemma 3, it gets

$$\frac{k_1^2}{k_2}\Psi_3\left(t\right)\tilde{x}^T\left(H\Xi\otimes I_r\right)\tilde{x} = \frac{k_1^2}{2k_2}\Psi_3\left(t\right)\tilde{x}^T\left(Q\otimes I_r\right)\tilde{x},$$
$$k_2\Psi_3\left(t\right)\tilde{v}^T\left(H\Xi\otimes I_r\right)\tilde{v} = \frac{k_2}{2}\Psi_3\left(t\right)\tilde{v}^T\left(Q\otimes I_r\right)\tilde{v}.$$

(31)

The $\dot{V}_2$ can be simplified as follows

$$\dot{V}_2 = \frac{k_1}{\rho}\Psi_3\left(t\right)\tilde{x}^T\left(Q\otimes I_r\right)\tilde{x} - \frac{k_1^2}{k_2}\Psi_3\left(t\right)\tilde{x}^T\left(H\Xi\otimes I_r\right)\tilde{x} + \frac{k_1}{k_2}\Psi_3\left(t\right)\tilde{v}^T\left(\Xi\otimes I_r\right)\tilde{v}$$
$$-k_2\Psi_3\left(t\right)\tilde{v}^T\left(H\Xi\otimes I_r\right)\tilde{v} + \tilde{x}^T\left(k_1\Psi_3\left(t\right)\left(Q\otimes I_r\right) + \frac{k_1}{k_2\rho}\Psi_3\left(t\right)\left(\Xi\otimes I_r\right)\right)\tilde{v}$$
$$-k_1\Psi_3\left(t\right)\tilde{x}^T\left(Q\otimes I_r\right)\tilde{v}$$
$$\le \left[\left(\frac{k_1}{\rho} - \frac{k_1^2}{2k_2}\right)\frac{\lambda_{\max}\left(Q\right)}{\lambda_{\min}\left(Q\right)} + \frac{k_1}{2k_2\rho}\frac{\lambda_{\max}\left(\Xi\right)}{\lambda_{\min}\left(Q\right)}\right]\Psi_3\left(t\right)\tilde{x}^T\left(Q\otimes I_r\right)\tilde{x}$$
$$+\left[\frac{k_1}{k_2}\frac{\lambda_{\max}\left(\Xi\right)}{\lambda_{\min}\left(Q\right)}\left(1 + \frac{1}{\rho}\right) - \frac{k_2}{2}\right]\Psi_3\left(t\right)\tilde{v}^T\left(Q\otimes I_r\right)\tilde{v}.$$

(32)

Combining (28), (29), (30) and (32), it gets

$$
\begin{aligned}
\dot{V}_2 &\leq \left[\left(\frac{k_1}{\rho} - \frac{k_1^2}{2k_2}\right)\frac{\lambda_{\max}(Q)}{\lambda_{\min}(Q)} + \frac{k_1}{2k_2\rho}\upsilon_1\right]\Psi_3(t)\,\tilde{x}^T\left(Q\otimes I_r\right)\tilde{x} \\
&\quad + \left[\frac{k_1}{k_2}\upsilon_1\left(1+\frac{1}{\rho}\right) - \frac{k_2}{2}\right]\Psi_3(t)\,\tilde{v}^T\left(Q\otimes I_r\right)\tilde{v} \\
&\quad + \left[\upsilon_2\left(\frac{k_1 Tc_x}{k_2\rho} + \frac{Tlc_x}{\rho} + \frac{k_1 lc_v}{k_2}\right) + \frac{lk_1}{2k_2} + \frac{(k_1+k_2)\lambda_{\max}(Q)}{8k_2 l\lambda_{\min}(Q)}\alpha^2 k_1^2\right]\Psi_3(t)\,\tilde{x}^T\left(Q\otimes I_r\right)\tilde{x} \\
&\quad + \left[\upsilon_2\left(c_v + \frac{Tc_x}{\rho 4l} + \frac{k_1 c_v}{k_2 4l}\right) + \frac{1}{4l} + \frac{(k_1+k_2)\lambda_{\max}(Q)}{8k_2 l\lambda_{\min}(Q)}\alpha^2 k_2^2\right]\Psi_3(t)\,\tilde{v}^T\left(Q\otimes I_r\right)\tilde{v} \\
&\quad + \lambda_{\max}(Q)\,\Psi_3(t)\,\gamma_i^2\psi^{-4b}(t) \\
&\leq (\Delta_1+\Delta_3)\,\Psi_3(t)\,\tilde{x}^T\left(Q\otimes I_r\right)\tilde{x} + (\Delta_2+\Delta_4)\,\Psi_3(t)\,\tilde{v}^T\left(Q\otimes I_r\right)\tilde{v} + \Delta_5\psi^{-4b}(t).
\end{aligned}
\tag{33}
$$

Let $Z = \begin{bmatrix} (\Delta_1+\Delta_3)\,Q & 0 \\ 0 & (\Delta_2+\Delta_4)\,Q \end{bmatrix}$, then

$$
\dot{V}_2 \leq \frac{\lambda_{\max}(Z)}{\lambda_{\min}(\Omega)}\Psi_3(t)\,V + \Delta_5\psi^{-4b}(t),\ t\in\left[t_0,T_3\right),
\tag{34}
$$

and

$$
\dot{V}_2 \leq \frac{\lambda_{\max}(Z)}{\lambda_{\min}(\Omega)}\Psi_3(t)\,V, t\in\left[T_3,\infty\right).
\tag{34}
$$

Then, by Lemma 5, the system is prescribed-time stable. The following solution can be obtained

$$
\begin{cases}
\dot{V}_2 \leq \frac{\lambda_{\max}(Z)}{\lambda_{\min}(\Omega)}\Psi_3(t)\,V_2 + \Delta_5\mu^{-4b}(t), & t\in\left[t_0,t_0+T_3\right), \\
\dot{V}_2 \leq \frac{\lambda_{\max}(Z)}{\lambda_{\min}(\Omega)}\Psi_3(t)\,V_2, & t\in\left[t_0+T_3,\infty\right).
\end{cases}
$$

That is, the lag bipartite consensus error $\bar{x}_i(t)$ and $\bar{v}_i(t)$ converges to zero within $T_3$. Then, the PTLBC of the MASs and can be achieved.

**Theorem 3.** Considering the MASs (8) and (9) with the PTLBC control protocol, the Zeno behaviour can be excluded.

**Proof.** Taking the derivative of $e(t)$, it has

$$
\begin{aligned}
\|\dot{e}(t)\| &= \|\dot{\eta}(t)\| = \|k_1\dot{\tilde{x}}_i(t) + k_2\dot{\tilde{v}}_i(t)\| \\
&= \|k_1\left(\dot{\Psi}(t)\tilde{x}_i(t) + \Psi(t)\tilde{v}_i(t)\right) + k_2\left(F(t,\tilde{x}_i(t),\tilde{v}_i(t)) + k_1\left(L\otimes I_N\right)\tilde{x}_i(t_k^i) + k_2\left(L\otimes I_N\right)\tilde{v}_i(t_k^i)\right)\| \\
&\leq \|\left(k_1\dot{\Psi}(t) + k_2 c_x\right)\tilde{x}_i(t)\| + \|\left(k_1\Psi(t) + k_2 c_v\right)\tilde{v}_i(t)\| + \|\Xi(t_k^i)\|.
\end{aligned}
$$

Let $\Delta = \max\left\{\left(k_1\dot{\Psi}(t) + k_2 c_x\right),\left(k_1\Psi(t) + k_2 c_v\right)\right\}$, and then

$$
\begin{aligned}
\|\dot{e}(t)\| &\leq \Delta\left(\|\tilde{x}_i(t)\| + \|\tilde{v}_i(t)\|\right) + \|\Xi(t_k^i)\| \\
&\leq 2\Delta\|\delta(t)\| + \|\Xi(t_k^i)\|.
\end{aligned}
$$

According to (34), it gets

$$
V(t) < \left[V(t_0) + \frac{\Delta_5 T}{h\left(\frac{\lambda_{\min}(Z)}{\lambda_{\max}(\Omega)} - 4b\right) - 1}\right]\psi^{-4b}(t).
\tag{36}
$$

Then, according to (26), it obtains

$$\lambda_{\min}(\Omega)\,\delta^T\delta < V(t) < \left[ V(t_0) + \frac{\Delta_5 T}{h\left(\frac{\lambda_{\min}(Z)}{\lambda_{\max}(\Omega)} - 4b\right) - 1} \right] \psi^{-4b}(t).$$

(37)

Hence

$$\|\delta(t)\|^2 \leq \frac{1}{\lambda_{\min}(\Omega)} \left[ V(t_0) + \frac{\Delta_5 T}{h\left(\frac{\lambda_{\min}(Z)}{\lambda_{\max}(\Omega)} - 4b\right) - 1} \right] \psi^{-4b}(t).$$

(38)

Obviously

$$\|\delta(t)\| \leq \sqrt{\frac{1}{\lambda_{\min}(\Omega)} \left[ V(t_0) + \frac{\Delta_5 T}{h\left(\frac{\lambda_{\min}(Z)}{\lambda_{\max}(\Omega)} - 4b\right) - 1} \right]} \,\psi^{-2b}(t).$$

(39)

And then

$$\|\dot{e}(t)\| \leq 2\Delta \sqrt{\frac{1}{\lambda_{\min}(\Omega)} \left[ V(t_0) + \frac{\Delta_5 T}{h\left(\frac{\lambda_{\min}(Z)}{\lambda_{\max}(\Omega)} - 4b\right) - 1} \right]} \,\psi^{-2b}(t) + \|\Xi(t_k^i)\|.$$

(40)

Considering $e_i(t_k^i) = 0$, it obtains

$$\|e(t)\| \leq \int_{t_k^i}^t 2\Delta \sqrt{\frac{1}{\lambda_{\min}(\Omega)} \left[ V(t_0) + \frac{\Delta_5 T}{h\left(\frac{\lambda_{\min}(Z)}{\lambda_{\max}(\Omega)} - 4b\right) - 1} \right]} \,\psi^{-2b}(\tau)\,d\tau + \int_{t_k^i}^t \|\Xi(t_k^i)\|\,d\tau.$$

(41)

At trigger instant $t_{k+1}^i$, it follows

$$\|e(t_{k+1}^i)\| \leq -\frac{2\Delta \sqrt{\frac{1}{\lambda_{\min}(\Omega)} \left[ V(t_0) + \frac{\Delta_5 T}{h\left(\frac{\lambda_{\min}(Z)}{\lambda_{\max}(\Omega)} - 4b\right) - 1} \right]} \,\psi^{-2b}(T+t_0)^{1-2\rho b}}{2\rho b} \times \left( (T+t_0-t_{k+1}^i)^{2\rho b} - (T+t_0-t_k^i)^{2\rho b} \right) + \|\Xi(t_k^i)\|\,(t_{k+1}^i - t_k^i)$$

(42)

According to (18), at trigger instant $t_{k+1}^i$

$$\|e_i(t_{k+1}^i)\| \geq \alpha\|\eta_i(t_{k+1}^i)\| + \gamma_i \psi^{-2b}(t_{k+1}^i),$$

(43)

combining with (42) and (43), it follows

$$\gamma_i \psi^{-2b}(t_{k+1}^i) \leq -\frac{2\Delta \sqrt{\frac{1}{\lambda_{\min}(\Omega)} \left[ V(t_0) + \frac{\Delta_5 T}{h\left(\frac{\lambda_{\min}(Z)}{\lambda_{\max}(\Omega)} - 4b\right) - 1} \right]} \,\psi^{-2b}(T+t_0)^{1-2\rho b}}{2hb} \times \left( (T+t_0-t_{k+1}^i)^{2\rho b} - (T+t_0-t_k^i)^{2\rho b} \right) + \|\Xi(t_k^i)\|\,(t_{k+1}^i - t_k^i)$$

(44)

Next, using reduction to absurdity, it will be proved that the trigger interval $t_{k+1}^i - t_k^i \neq 0$. Assume $t_{k+1}^i - t_k^i = 0$, it can be deduced from that $0 < \gamma_i \leq 0$, which leads to a contradiction. Hence, the assumption does not hold, that is, the proposed event-triggered mechanism excludes Zeno behaviour.

## 4. Numerical results

The simulation results of nonlinear MASs (8) and (9) under the PTLBC control protocol (19) are given in the section.

In S1 Fig, the nonlinear MASs with one leader and six followers is presented. The solid line indicates cooperation relationships, and the dashed line indicates the competitive relationships of the followers. The followers are partitioned into two subsets $\mathcal{V}_1 \in \{v_1, v_2, v_3\}$ and $\mathcal{V}_2 \in \{v_4, v_5, v_6\}$, which means that $\theta = \text{diag}\,(1, 1, 1, -1, -1, -1)$. The Laplacian matrix $L$ is below.

$$L = \begin{bmatrix} 1 & 0 & -1 & 0 & 0 & 0 \\ 0 & 1 & -1 & 0 & 0 & 0 \\ -1 & -1 & 5 & 1 & 1 & 1 \\ 0 & 0 & 1 & 3 & -1 & -1 \\ 0 & 0 & 1 & -1 & 2 & 0 \\ 0 & 0 & 1 & -1 & 0 & 2 \end{bmatrix}.$$

Obviously, $B = \text{diag}\,(1, 1, 1, 0, 0, 0)$. For the leader, the control input is expressed as $u_0 = 0.1 * sin\,(t)$, and the nonlinear part is expressed as $g\,(t, x_0\,(t), v_0\,(t)) = 0.2v_0\,(t) \sin\,(x_0\,(t))$. The nonlinear parts of the followers are expressed as $g\,(t, x_i\,(t), v_i\,(t)) = 0.2v_i\,(t) \sin\,(x_i\,(t))$, $i = 1, 2, 3, 4, 5, 6$. Obviously, we can verify that Assumption 2 is satisfied with $c_x = 0.03$ and $c_v = 0.2$. Set $t_0 = 0s$, $T = 2s$ and $\tau = 1s$. The conditions (25) of Theorem 2 are satisfied when we choose $k_1 = 7$, $k_2 = 6.23$. The initial states values are given by $x_0\,(0) = 1$, $v_0\,(0) = 1$, $x\,(0) = \begin{bmatrix} -5 & -4 & 7 & 8 & 2 & 5 \end{bmatrix}^T$ and $v\,(0) = \begin{bmatrix} -3.5 & -0.5 & 2.5 & 1.5 & 5 & 1 \end{bmatrix}^T$ respectively. Setting the parameters $\alpha = 0.65$, $\gamma_i = \begin{bmatrix} 0.25 & 0.25 & 0.25 & 0.25 & 0.25 & 0.25 \end{bmatrix}$, $\rho = 1$ and $b = 0.25$ in the event-triggering function, the function is given as follows

$$h_i\,(t) = \|e_i\,(t)\| - 0.65\|\eta_i\,(t)\| - \gamma_i \psi^{-0.5}\,(t).$$

The initial values $\vartheta_{xi}$ and $\vartheta_{vi}$ of the PTDO are set to $\begin{bmatrix} 4, 0, 5, -5, -1, -4 \end{bmatrix}$ and $\begin{bmatrix} 5, -5, 3, 3, 4, -5 \end{bmatrix}$, respectively. S2 Fig shows that $\overline{\vartheta}_{xi}$ converges to 0 within the arbitrary prescribed time $2s$, and $\overline{\vartheta}_{vi}$ converges to 0 within the arbitrary prescribed time $4s$, with the lag time $1s$. This demonstrates that the PTDO is effective.

S3 Fig A and S3 Fig B provide the detailed variation curves of the position and velocity for all agents. It indicates that LBC is realized within the prescribed time $6s$, with the lag time $1s$.

S4 Fig presents the event intervals for follower, which confirms that Zeno behaviour does not occur.

## 5. Conclusion

The paper has researched the PTLBC control problem of nonlinear MASs. The PTDO has been devised for followers to acquire the states of the leader, and the event-triggered mechanism has been devised to save communication consumption. Via the PTDO and the event-triggered mechanism, the PTLBC control protocol has been designed. Utilizing Lyapunov stability analysis method, it has been rigorously proven that the nonlinear MASs (8) and (9) can achieve PTLBC. Furthermore, the Zeno behaviour of the controllers has been excluded. And then, the theoretical results were verified through the simulation example. We will further explore the problem subject to cyber attacks in our future work.

## Supporting information

**S1 Fig. The connected topology of the nonlinear MASs.**
(TIF)

**S2 Fig. The observer errors of followers.**
(TIF)

**S3 Fig. The states variation curves of all agents.**
(TIF)

**S4 Fig. Triggering instant sequences of followers.**
(TIF)

**S1 File. Data for Numerical results.**
(ZIP)

## Author contributions

**Conceptualization:** Tao Li.

**Funding acquisition:** Tao Li, Xiaowen Zhao.

**Investigation:** Xiaowen Zhao, Le Yan.

**Methodology:** Jialong Tian, Tao Li, Yuanmei Wang.

**Software:** Jialong Tian, Haiyang Hu.

**Supervision:** Tao Li.

**Validation:** Yuanmei Wang, Haiyang Hu.

**Writing – original draft:** Jialong Tian.

**Writing – review & editing:** Le Yan.

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
