## [Decision Letter · Decision Letter 0]

24 Mar 2026

PONE-D-26-08045Observer-based prescribed-time lag bipartite consensus of nonlinear multi-agent systems under event-triggered mechanismPLOS One

Dear Dr. Li,

Thank you for submitting your manuscript to PLOS ONE. After careful consideration, we feel that it has merit but does not fully meet PLOS ONE’s publication criteria as it currently stands. Therefore, we invite you to submit a revised version of the manuscript that addresses the points raised during the review process.

As the corresponding author, your ORCID iD is verified in the submission system and will appear in the published article. PLOS supports the use of ORCID, and we encourage all coauthors to register for an ORCID iD and use it as well. Please encourage your coauthors to verify their ORCID iD within the submission system before final acceptance, as unverified ORCID iDs will not appear in the published article. Only the individual author can complete the verification step; PLOS staff cannot verify ORCID iDs on behalf of authors.

We look forward to receiving your revised manuscript.

Kind regards,

Hongru Ren

Academic Editor

PLOS One

**Journal Requirements:**

1. When submitting your revision, we need you to address these additional requirements. Please ensure that your manuscript meets PLOS ONE's style requirements, including those for file naming. The PLOS ONE style templates can be found at https://journals.plos.org/plosone/s/file?id=wjVg/PLOSOne_formatting_sample_main_body.pdf and https://journals.plos.org/plosone/s/file?id=ba62/PLOSOne_formatting_sample_title_authors_affiliations.pdf 2. Please note that PLOS One has specific guidelines on code sharing for submissions in which author-generated code underpins the findings in the manuscript. In these cases, we expect all author-generated code to be made available without restrictions upon publication of the work. Please review our guidelines at https://journals.plos.org/plosone/s/materials-and-software-sharing#loc-sharing-code and ensure that your code is shared in a way that follows best practice and facilitates reproducibility and reuse. 3. Thank you for stating the following financial disclosure: Tao Li received the funding of the National Natural Science Foundation of China under Grant number 62473135, Xiaowen Zhao received the funding of the National Natural Science Foundation of China under Grant number 62173121, Le Yan received the Natural Science Foundation of Hubei Province of China under Grant number 2024AFB812.   Please state what role the funders took in the study.  If the funders had no role, please state: "The funders had no role in study design, data collection and analysis, decision to publish, or preparation of the manuscript." If this statement is not correct you must amend it as needed. Please include this amended Role of Funder statement in your cover letter; we will change the online submission form on your behalf. 4. Thank you for stating the following in the Acknowledgments Section of your manuscript: This work is supported by the National Natural Science Foundation of China under Grants 62473135 and 62173121, in part by the Natural Science Foundation of Hubei Province of China under Grant 2024AFB812. We note that you have provided funding information that is not currently declared in your Funding Statement. However, funding information should not appear in the Acknowledgments section or other areas of your manuscript. We will only publish funding information present in the Funding Statement section of the online submission form. Please remove any funding-related text from the manuscript and let us know how you would like to update your Funding Statement. Currently, your Funding Statement reads as follows: Tao Li received the funding of the National Natural Science Foundation of China under Grant number 62473135, Xiaowen Zhao received the funding of the National Natural Science Foundation of China under Grant number 62173121, Le Yan received the Natural Science Foundation of Hubei Province of China under Grant number 2024AFB812.  Please include your amended statements within your cover letter; we will change the online submission form on your behalf. 5. We note that your Data Availability Statement is currently as follows: All relevant data are within the manuscript and its Supporting Information files. Please confirm at this time whether or not your submission contains all raw data required to replicate the results of your study. Authors must share the “minimal data set” for their submission. PLOS defines the minimal data set to consist of the data required to replicate all study findings reported in the article, as well as related metadata and methods (https://journals.plos.org/plosone/s/data-availability#loc-minimal-data-set-definition). For example, authors should submit the following data: - The values behind the means, standard deviations and other measures reported;- The values used to build graphs;- The points extracted from images for analysis. Authors do not need to submit their entire data set if only a portion of the data was used in the reported study. If your submission does not contain these data, please either upload them as Supporting Information files or deposit them to a stable, public repository and provide us with the relevant URLs, DOIs, or accession numbers. For a list of recommended repositories, please see https://journals.plos.org/plosone/s/recommended-repositories. If there are ethical or legal restrictions on sharing a de-identified data set, please explain them in detail (e.g., data contain potentially sensitive information, data are owned by a third-party organization, etc.) and who has imposed them (e.g., an ethics committee). Please also provide contact information for a data access committee, ethics committee, or other institutional body to which data requests may be sent. If data are owned by a third party, please indicate how others may request data access. 6. Please upload a new copy of Figures 2, 3, and 4, as the detail is not clear. Please follow the link for more information:  https://journals.plos.org/plosone/s/figures 7. If the reviewer comments include a recommendation to cite specific previously published works, please review and evaluate these publications to determine whether they are relevant and should be cited. There is no requirement to cite these works unless the editor has indicated otherwise.

**Additional Editor Comments:**

In light of the reviewers’ comments, the manuscript is considered to have potential for publication, provided that appropriate and necessary revisions are made. The authors are kindly requested to carefully address these comments and further improve the manuscript accordingly.

Reviewers' comments:

Reviewer's Responses to Questions

**Comments to the Author**

1. Is the manuscript technically sound, and do the data support the conclusions?

Reviewer #1: Yes

Reviewer #2: Yes

2. Has the statistical analysis been performed appropriately and rigorously?

Reviewer #1: Yes

Reviewer #2: Yes

3. Have the authors made all data underlying the findings in their manuscript fully available?

Reviewer #1: No

Reviewer #2: No

4. Is the manuscript presented in an intelligible fashion and written in standard English?

Reviewer #1: Yes

Reviewer #2: Yes

5. Review Comments to the Author

**Reviewer #1:** This manuscript studies the observer-based prescribed-time lag bipartite consensus problem for nonlinear multi-agent systems under an event-triggered mechanism. The topic is relevant, and the integration of prescribed-time observer design, event-triggered communication, and lag bipartite consensus is potentially interesting. However, the following comments must be considered:

1. The notation system and several basic definitions need substantial revision.

2. Theorem 2 invokes Assumptions 1, 2, and 3, but Assumption 3 is not explicitly stated in the provided preliminaries.

3. The prescribed-time design and the time parameters T, T_1, T_2, and T_3 should be clarified much more carefully.

4. The main proofs, especially for Theorems 1 and 2, rely on several matrix inequalities and positive-definiteness claims, but the derivations are presented too briefly. In particular, the manuscript needs to justify more explicitly why the chosen Lyapunov functions are positive definite under the proposed parameter conditions, and how the bounds are transformed step by step.

5. The Introduction would benefit from citing and discussing several references relevant to the research methodology, which could further enrich the background on nonlinear control and event-triggered control, such as 10.1016/j.automatica.2024.111511, 10.1016/j.automatica.2022.110792, and 10.1109/TIE.2026.3654623.

**Reviewer #2:** This paper aims to address the prescribed-time lag bipartite consensus control problem for nonlinear multi-agent systems. Generally, this is a well written and organized article with interesting theoretical results proposed. However, some concerns have to be addressed for better qualification:

The contribution should be stated by comparing with the existing results, therefore, the novelty should be further improved and highlighted.

The reasonability and references of assumptions should be given.

Some latest results about finite-time control should be introduced such as tuning function-based light computational adaptive fixed-time control for overhead cranes with multiple uncertainties, security-driven adaptive iterative learning formation control for multiagent systems etc., to make the review complete.

Fig. 4 is unclear, please revise it.

Some comparison results should be shown to explain the novelty of the proposed method.

The future work should be added to the conclusion.

The linguistic quality or grammar problems need improvement, such as "\beta are the observer gains" etc. The authors should pay attention to these problems.

6. PLOS authors have the option to publish the peer review history of their article (what does this mean?). If published, this will include your full peer review and any attached files.

**Do you want your identity to be public for this peer review?** For information about this choice, including consent withdrawal, please see our Privacy Policy.

Reviewer #1: No

Reviewer #2: No

---

## [Author Response · Author response to Decision Letter 1]

23 Apr 2026

The authors wish to thank the Editor and the anonymous reviewers very much for their valuable comments and suggestions, which leads to significant improvement of the quality and presentation of the manuscript. In the attachment "Response to Reviewers.docx" , we have given the detailed explanation how the editor’s and reviewer' comments have been taken into account in the revision.

---

## [Decision Letter · Decision Letter 1]

6 May 2026

Observer-based prescribed-time lag bipartite consensus of nonlinear multi-agent systems under event-triggered mechanism

PONE-D-26-08045R1

Dear Dr. Li,

We’re pleased to inform you that your manuscript has been judged scientifically suitable for publication and will be formally accepted for publication once it meets all outstanding technical requirements.

Kind regards,

Hongru Ren

Academic Editor

PLOS One

Additional Editor Comments (optional):

The revised manuscript has been re-reviewed by multiple reviewers and has received favorable evaluations and unanimous approval from all of them. After carefully assessing the reviewers’ comments and the overall academic quality of the manuscript, I fully agree with the reviewers’ judgments and recommendations.

Reviewers' comments:

Reviewer's Responses to Questions

**Comments to the Author**

1. If the authors have adequately addressed your comments raised in a previous round of review and you feel that this manuscript is now acceptable for publication, you may indicate that here to bypass the “Comments to the Author” section, enter your conflict of interest statement in the “Confidential to Editor” section, and submit your "Accept" recommendation.

Reviewer #1: All comments have been addressed

Reviewer #2: All comments have been addressed

2. Is the manuscript technically sound, and do the data support the conclusions?

Reviewer #1: Yes

Reviewer #2: Yes

3. Has the statistical analysis been performed appropriately and rigorously? 

Reviewer #1: Yes

Reviewer #2: Yes

4. Have the authors made all data underlying the findings in their manuscript fully available?

Reviewer #1: Yes

Reviewer #2: Yes

5. Is the manuscript presented in an intelligible fashion and written in standard English?

Reviewer #1: Yes

Reviewer #2: Yes

6. Review Comments to the Author

Reviewer #1: The authors have addressed all my concerns. I think this paper can be accept now. Good luck to the authors.

Reviewer #2: This paper has been revised well. I have no further comments. As a result, I suggest that it can be accepted.

7. PLOS authors have the option to publish the peer review history of their article (what does this mean?). If published, this will include your full peer review and any attached files.

Reviewer #1: No

Reviewer #2: No
